# OpenReview forum: "The Devil behind the mask: An emergent safety vulnerability of Diffusion LLMs"
_ICLR.cc/2026/Conference — ICLR 2026 Poster_

### Official Review · Reviewer_Jo4s · 2025-10-30

**Soundness:** 2
**Presentation:** 1
**Contribution:** 2
**Rating:** 2
**Confidence:** 4

**Summary:**

This paper investigates jailbreak vulnerability in Diffusion-based Large Language Models (dLLMs), arguing that their unique bidirectional architecture is inadequately protected by existing safety alignment methods. To exploit this, the authors propose DIJA, an interleaved, masked jailbreak attack that achieves a high success rate against dLLMs and remains robust when tested against state-of-the-art defenses like Robust Prompt Optimization (RPO).

**Strengths:**

* The paper identifies and explores an emergent safety vulnerability specific to dLLMs.

**Weaknesses:**

## Major

1. **Missing Ablation Study:** The DIJA method is a composite of several ideas: prompt refinement, different masking patterns (block-wise, fine-grained, progressive), and benign separator insertion. The paper fails to isolate the contribution of each component. Without an ablation study, it is impossible to determine whether the high ASR is due to the novel masking patterns, or simply the use of a more powerful Refinement LLM for prompt generation. The current analysis exploring the number of masked tokens is a hyperparameter sweep, not a methodological ablation.
2. **Analysis of Off-the-shelf Baseline (especially ReNeLLM):** The authors' results show that ReNeLLM (a simple, off-the-shelf attack) is surprisingly competitive, even outperforming DIJA on the LLaDA-Instruct target model on the JailbreakBench dataset (ReNeLLM at 96.0% ASR-e vs. DIJA at 81.0% ASR-e). If a generic attack performs equally or better than the proposed, highly-specialized method, it questions the core claim of DIJA's necessity for attacking dLLMs. The authors must address this discrepancy. Somehow, this is also related to weakness 1-- as we don't really understand what part of the approach is actually contributing to the effectiveness of the attack

3. **Methodological Ambiguity:** The methodology is presented at times in a unnecessarily complex way. Also, the role of the initial prompt 'a' in Algorithm 1 is confusingly presented. While it is the source of the harmful content, its relationship to the final attack prompt is unclear to me: is it a successfull JB? Against what victim model? etc. The paper should clearly state that 'a' is a seed/template for the attack, and the LLM $\mathcal{L}$ is the attacker LLM that generates the final interleaved attack prompt $p_i$ used against the target dLLM $\mathcal{D}$. This two-stage process needs clearer articulation.

## Minor

* **Equation (2) Notation:** The variable $\Phi$ in Equation (2) is not explicitly defined. While context suggests it represents the set of possible masks or masking operations, this must be formally stated for a technical paper.
* **Overclaiming of Effectiveness:** The claim on page 7 regarding the high ASR being "rarely observed on autoregressive models" is an overclaim. Given that several recent, powerful attacks (like GCG variants or certain adaptive attacks) routinely achieve near-100% ASR on state-of-the-art autoregressive models, this statement should be tempered or removed. The true novelty is the method of attack (masking), not necessarily the ASR magnitude.
* **Clarity and Readability:** Section 3.2.2: The section should not start with "Specifically,".Section 3.2.1 Motivation: The section is titled "Motivation" but contains minimal motivational text, primarily diving straight into the technical details.
* **Figure Clarity:** Figures 1, 2, 4, and 6 are unclear and difficult to read, especially in a double-column format. The authors should improve image quality and legend clarity for the final version.

**Questions:**

I suggets  the authors should:

- Implement a full ablation study to quantify the contribution of masking vs. prompt refinement vs. benign separators.

- Explain the ReNeLLM discrepancy, perhaps with a discussion on why dLLMs' bidirectional nature makes them susceptible to both simple and complex attacks.

---

> ### Author Response · Authors · 2025-11-21
> **Response to Reviewer Jo4s (part 1)**
>
> Dear Reviewer Jo4s,
>
> Thank you very much for taking the time to review our paper and for providing your feedback. Below, I will address your questions one by one.
>
> > **W1.** Missing Ablation Study: The DIJA method is a composite of several ideas: prompt refinement, different masking patterns (block-wise, fine-grained, progressive), and benign separator insertion. The paper fails to isolate the contribution of each component. Without an ablation study, it is impossible to determine whether the high ASR is due to the novel masking patterns, or simply the use of a more powerful Refinement LLM for prompt generation. The current analysis exploring the number of masked tokens is a hyperparameter sweep, not a methodological ablation. \
> > **Q1.** Implement a full ablation study to quantify the contribution of masking vs. prompt refinement vs. benign separators.
>
> **RW1&Q1:** We sincerely thank the reviewer for this valuable suggestion. Conducting a detailed ablation study on the components of the DIJA framework is indeed essential to reveal the essence of our method.
>
> Specifically, we conducted the following ablation experiments on HarmBench using LLaDA-8B-Instruct:
> 1. **Ablation on Prompt Refinement (w/o Refinement LLM):** To prove that the effectiveness of DIJA does not stem from the rewriting capability of the auxiliary LLM, we completely excluded the LLM and the diverse few-shot demonstrations in this setting. We retained the interleaved mask-text structure but used a fixed, heuristic template for benign separators (e.g., "First, [MASK]... Second, [MASK]... Third, [MASK]..."), keeping the number of mask tokens identical to the standard setting.
> 2. **Ablation on Masking (w/o Masking Mechanism):** Based on the full DIJA framework, we removed all masked regions from the generated interleaved prompts, retaining only the vanilla harmful query and the benign separators. This reverts the model's inference to standard generation (as shown in Eq. 1-4 of the main text), testing whether the semantic guidance alone is sufficient.
> 3. **Ablation on Benign Separators (w/o Separators):** Based on the full DIJA framework, we removed all benign separators from the generated interleaved mask-text prompts, retaining only the vanilla harmful query and the mask tokens (i.e., Query + [MASK]...).
>
> *Note: Ablations 2 and 3 essentially disrupt the critical **interleaved mask-text prompt pattern**.*
>
> **Results in Ablation Study**
>
> | Metric | **w/o Refinement LLM** | **w/o Masking** | **w/o Separators** | **DIJA (Full)** |
> | :---: | :---: | :---: | :---: | :---: |
> | **ASR-k** | 94.5% | 47.5% | 53.5% | **96.3%** |
> | **ASR-e** | 54.8% | 14.0% | 16.8% | **55.5%** |
> | **HS** | 4.0 | 2.9 | 3.1 | **4.1** |
>
> The experimental results show that the interleaved mask–text prompt is the key to the success of the attack. Even when we remove the LLM used for prompt refinement in the DIJA framework, as long as the interleaved mask–text structure is preserved, the attack remains highly effective. This further highlights the fundamental difference between our method and prior rewriting-based or disguise-type jailbreak attacks: our approach is grounded in the intrinsic properties of dLLMs. By uncovering a new class of vulnerabilities specific to these models, we design an attack framework that is fundamentally aligned with their underlying mechanisms.
>
> **Once again, thank you for your valuable suggestions. We have added these experimental results and the corresponding discussion to Appendix B.5 in the revision version, and we hope this addresses your concerns.**
>
> > **W2.** Analysis of Off-the-shelf Baseline (especially ReNeLLM): The authors' results show that ReNeLLM (a simple, off-the-shelf attack) is surprisingly competitive, even outperforming DIJA on the LLaDA-Instruct target model on the JailbreakBench dataset (ReNeLLM at 96.0% ASR-e vs. DIJA at 81.0% ASR-e). If a generic attack performs equally or better than the proposed, highly-specialized method, it questions the core claim of DIJA's necessity for attacking dLLMs. The authors must address this discrepancy. Somehow, this is also related to weakness 1-- as we don't really understand what part of the approach is actually contributing to the effectiveness of the attack \
> > **Q2.** Explain the ReNeLLM discrepancy, perhaps with a discussion on why dLLMs' bidirectional nature makes them susceptible to both simple and complex attacks.
>
> **RW2 & Q2:** Thank you for this detailed comparison. We appreciate the opportunity to clarify the positioning of DIJA relative to baseline ReNeLLM.
>
> (continued in our next response)

---

> ### Author Response · Authors · 2025-11-21
> **Response to Reviewer Jo4s (part 2)**
>
> **1. Clarification on Table 2 Data:**
> Regarding the performance of ReNeLLM on LLaDA-Instruct, we would like to respectfully clarify the data presented in Table 2. It appears there might be a slight confusion regarding the metrics:
> *   The figure of **96.0%** corresponds to ReNeLLM's **Keyword-based ASR (ASR-k)**.
> *   For the **Evaluator-based ASR (ASR-e)**, which you referred to, ReNeLLM achieves **80.0%**, whereas our DIJA achieves **81.0%**.
> Therefore, DIJA actually outperforms ReNeLLM on this metric, rather than underperforming. We apologize if the table layout caused any ambiguity.
>
> **2. Limitations of Keyword-based ASR (ASR-k):**
> It can be observed that ReNeLLM achieves competitive results on ASR-k. While this metric is widely used in trustworthy AI research, we wish to highlight its inherent limitations. ASR-k determines "success" solely by the absence of refusal keywords. However, in practice, models often generate **irrelevant, nonsensical, or evasive content** that lacks refusal keywords but also fails to provide the requested harmful information. Such cases are counted as "successes" by ASR-k but have little practical value in a real-world attack scenario.
> Therefore, we strongly advocate for comparing methods based on **ASR-e** and **Harmfulness Score (HS)**, which evaluate the *actual content* and semantic harmfulness of the response. Under these more rigorous metrics, the advantage of DIJA becomes more apparent.
>
> **3. The Necessity of DIJA (Critical Evidence):**
> Under these rigorous metrics (ASR-e), the necessity of DIJA is best demonstrated on victim models with stronger safety alignment, such as **Dream-Instruct** (see Table 2). In this setting, the distinction becomes stark:
> *   **ReNeLLM (Generic):** Its ASR-e drops drastically to **11.5%**, indicating that standard rewriting attacks are easily blocked by robustly aligned dLLMs.
> *   **DIJA:** It maintains a remarkably high ASR-e of **90.0%**.
>
> This **78.5% gap** definitively proves that generic off-the-shelf attacks are insufficient for assessing or securing robust dLLMs. DIJA is necessary because it exploits the **architectural vulnerability**, allowing it to bypass safeguards that successfully defend against semantic rewriting.
>
> **4. Mechanism Explanation:**
> Regarding your question on why dLLMs are susceptible to both:
> *  **ReNeLLM** is indeed a very interesting work. It succeeds on models like LLaDA primarily because these models have weaker initial alignment against semantic deception. However, it is important to note that the success of ReNeLLM largely relies on carefully designed, complex scenarios and model deception, rather than properties unique to dLLMs. This is evident from its sharply reduced effectiveness on models with stronger defenses, such as Dream, whereas our method continues to perform well.
> *   **DIJA** succeeds on both weak (LLaDA) and strong (Dream) dLLMs because it exploits the **fundamental bidirectional modeling and parallel decoding mechanism**. By enforcing contextual coherence via masks, DIJA overrides the model's refusal logic, a vulnerability intrinsic to the architecture that persists even when semantic alignment is strengthened.
>
> We hope our response addresses your concerns. Thank you once again!
>
> > **W3.** Methodological Ambiguity: The description is sometimes overly complex. The role of the initial prompt 'a' is unclear, it should be stated that 'a' is a seed for the attack, and the LLM generates the final attack prompt for the target dLLM. The two-stage process needs clearer explanation.
>
> **RW3:** We thank you for pointing out the ambiguity in our methodology presentation. We agree that the definition of `a` and the two-stage process can be articulated more clearly.
>
> **1. Clarification on `a`:**
> To clarify, `a` is simply the **vanilla harmful prompt** (e.g., "Write a script to hack a website") sourced directly from benchmarks like HarmBench. It serves as the **seed containing the harmful intent**. It is **not** necessarily a pre-existing successful jailbreak against any specific victim model.
>
> **2. Clarification on the Two-Stage Process:**
> *   **Stage 1 (Transformation):** The Attacker LLM ($\mathcal{L}$) takes this seed `a` and transforms it into an interleaved mask-text prompt ($\mathbf{p}_i$) via in-context learning.
> *   **Stage 2 (Attack):** This generated prompt $\mathbf{p}_i$ is then fed to the Target dLLM ($\mathcal{D}$) to induce harmful generation.
>
> In the revision, we have updated **Algorithm 1** and **Section 3.2** to reflect these clarifications:
> *   We renamed `Vanilla jailbreak prompt a` to **`Vanilla harmful prompt a`** to avoid confusion.
> *   We added explicit comments in the algorithm to label $\mathcal{L}$ as the **"Attacker LLM"** and $\mathcal{D}$ as the **"Target Victim dLLM"**.
> *   We rewrote the description to explicitly delineate this two-stage workflow as suggested.
>
> (continued in our next response)

---

> > ### Author Response · Authors · 2025-11-21
> > **Response to Reviewer Jo4s (part 3)**
> >
> > > **W4.** Equation (2) Notation: The variable $\Phi$
> >  in Equation (2) is not explicitly defined. While context suggests it represents the set of possible masks or masking operations, this must be formally stated for a technical paper.
> >
> > **RW4:** Thank you for catching this notational omission. The symbol $\phi$ in Equation (2) denotes the **model parameters of dLLM**. We apologize for not defining it explicitly in the original text. **In the revised **Section 3.1**, we have added the formal definition.**
> >
> > > **W5.** Overclaiming of Effectiveness: The claim on page 7 regarding the high ASR being "rarely observed on autoregressive models" is an overclaim. Given that several recent, powerful attacks (like GCG variants or certain adaptive attacks) routinely achieve near-100% ASR on state-of-the-art autoregressive models, this statement should be tempered or removed. The true novelty is the method of attack (masking), not necessarily the ASR magnitude.
> >
> > **RW5:** We appreciate you for pointing out this overstatement. We agree that recent advanced attacks on autoregressive models (e.g., GCG, PAIR) also achieve very high ASRs, making our original claim imprecise.
> >
> > Our intention was to highlight that DIJA achieves such high success rates **without** requiring computationally expensive gradient optimization (like GCG) or producing incoherent gibberish. The high ASR is achieved via simple, semantically fluent interleaved structures, which is the key distinction.
> >
> > **In the revision (Section 4.2), we have **removed the claim** that high ASR is "rarely observed on autoregressive models." Instead, we now emphasize that DIJA achieves state-of-the-art attack performance efficiently by exploiting the unique architectural vulnerability of dLLMs, rather than comparing the absolute ASR magnitude against all AR attacks.**
> >
> > > **W6.** Clarity and Readability: Section 3.2.2: The section should not start with "Specifically,".Section 3.2.1 Motivation: The section is titled "Motivation" but contains minimal motivational text, primarily diving straight into the technical details.
> >
> > **RW6:** We thank the reviewer for these constructive suggestions on clarity and structure. \
> > We have removed the abrupt transition "Specifically," and rewrote the opening sentence to be more direct and self-contained: *"We utilize a refinement language model to automatically construct..."* in the revised version. \
> > We agree that the original content of Section 3.2.1 focused more on the mathematical definition of the problem rather than the high-level motivation. To accurately reflect the content, we have **renamed the subsection to "Problem Formulation"** in the revision.
> >
> > > **W7.** Figure Clarity: Figures 1, 2, 4, and 6 are unclear and difficult to read, especially in a double-column format. The authors should improve image quality and legend clarity for the final version.
> >
> > **RW7:** Thank you for pointing out the readability issues in our figures. We will **redesign and update Figures 1, 2, 4, and 6** in the final manuscript.
> >
> > We hope our response addresses your concerns. If you have any further questions, please feel free to reach out to us. Thank you for your time and for the thoughtful feedback on our work.
> >
> > Best regards,
> >
> > Author Team

---

> > > ### Author Response · Authors · 2025-11-26
> > >
> > > Dear Reviewer Jo4s,
> > >
> > > Thank you once again for your valuable comments on our submission. As the discussion phase is approaching its end, we would like to kindly confirm whether we have sufficiently addressed all of your concerns (or at least part of them). Should there be any remaining questions or areas requiring further clarification, please do not hesitate to let us know. If you are satisfied with our responses, we would greatly appreciate your consideration in adjusting the evaluation scores accordingly. We sincerely look forward to your feedback.
> > >
> > > Best regards,
> > > Author Team

---

### Official Review · Reviewer_Jm6J · 2025-10-31

**Soundness:** 3
**Presentation:** 4
**Contribution:** 2
**Rating:** 4
**Confidence:** 4

**Summary:**

This paper identifies a critical safety vulnerability in diffusion-based large language models (dLLMs) arising from their core architectural features. The authors empirically validate this claim and propose DIJA, an automated jailbreak attack framework that constructs interleaved mask-text prompts to exploit these vulnerabilities: bidirectional modeling forces dLLMs to generate contextually consistent (even harmful) content for masked spans, while parallel decoding limits dynamic filtering of unsafe outputs.  Overall, this work highlights a previously overlooked threat surface in dLLM architectures and underscores the urgent need for rethinking safety alignment for this emerging class of language models.

**Strengths:**

- The paper features clear writing and a well-articulated motivation, making the research gap and significance intuitive to follow.
- The experimental design is comprehensive and robust, covering multiple representative general-purpose and code-oriented dLLMs, three major jailbreak benchmarks, and direct comparisons with state-of-the-art attack baselines.
- The paper proactively explores defensive mechanisms to add depth to safety analysis.

**Weaknesses:**

- The DIJA method appears too simple and just relies on a prompt template for generating interleaved mask-text prompts via in-context learning. Additionally, the paper provides no systematic analysis of the diversity of these mask-text adversarial prompts.
- The practical value of the research is limited due to the nascent stage of dLLM development. At present, dLLMs still suffer from noticeable gaps in training stability and the inference ecosystem, leaving few immediate landing scenarios.

**Questions:**

1. Since the authors use an existing LLM to generate harmful instructions for attacking dLLMs, how can the uncertainty of attack reproducibility be measured? For example, the mask-text prompts generated in multiple runs may exhibit significant differences in terms of structure or the number of mask tokens.
2. In lines 372-373, the authors claim: "This is because our method exposes the harmful intent in the prompt directly." As far as I know, role-playing attacks (such as AIM in the baseline) also incorporate the complete harmful intent into the prompt. How do you explain this inconsistency?

---

> ### Author Response · Authors · 2025-11-21
> **Response to Reviewer Jm6J (part 1)**
>
> Dear Reviewer Jm6J,
>
> Thank you very much for taking the time to review our paper and for providing your feedback. Below, I will address your questions one by one.
>
> > **W1.** Method Simplicity & Diversity Analysis
>
> **RW1:** Thank you for your valuable feedback. We would like to clarify the contributions of our method and address the concern regarding diversity.
> 1. **Simplicity as a Strength:** While DIJA's implementation is straightforward, we argue that this simplicity highlights the **fundamental nature** of the vulnerability in dLLMs. Unlike AR models that often require complex gradient-based optimization (GCG) or extensive iterative search (PAIR) to bypass safety guardrails, dLLMs can be compromised by simple interleaved patterns. This indicates that the vulnerability stems from the core architecture (bidirectional modeling and parallel decoding) rather than a lack of adversarial optimization effort. Our contribution lies in identifying and operationalizing this architecture-specific weakness.
>
> One important value of our work lies in drawing the community’s attention to these new risks and providing insights for the development and safety alignment of future dLLM training paradigms. We believe that the potential of dLLMs, enabled by parallel decoding, bidirectional context modeling, and the ability to iteratively remask and refine outputs, will continue to attract significant interest and drive downstream applications, as demonstrated by their recent adoption in both code generation [8], VLA [2], and medical tasks [3]. Therefore, it is crucial to address the security of dLLMs at an early stage.
>
> 2. As detailed in Section 3.2.2, we employ:
> - **Prompt Diversification:** We curate a diverse set of few-shot demonstrations that encompass various stylistic formats (e.g., step-by-step guides, Q&A sessions, code snippets). The primary objective of this diversification is to ensure that when transforming vanilla harmful queries from benchmarks like HarmBench into interleaved mask-text prompts, the resulting structures adaptively align with the inherent format of the original query. This alignment ensures that the constructed adversarial prompts remain linguistically natural and contextually fluent, thereby facilitating more effective bidirectional completion by the dLLM.
> - **Masking Pattern Selection:** We utilize three distinct strategies: Block-wise (redacting spans), Fine-grained (hiding keywords), and Progressive (step-by-step masking). To provide a systematic analysis of the diversity in masking patterns, we quantitatively evaluated the structural properties of our generated interleaved mask-text prompts. \
> **a.** Specifically, we report two key metrics: Mask Ratio, defined as the percentage of masked tokens relative to the total token count per prompt, and Average Mask Span Length, which measures the average length of contiguous masked segments within a prompt.
> These two metrics are critical indicators of structural diversity: Mask Ratio reflects the overall density of information concealment, distinguishing between prompts that require heavy generation versus those that only need minor infilling; meanwhile, Average Mask Span Length differentiates between fine-grained deletions (short spans) and block-wise redactions (long spans), verifying the presence of distinct masking strategies. \
> **b.** Taking the 400 samples generated from HarmBench as a representative set, our statistical analysis reveals a broad and healthy distribution. The Mask Ratio spans a wide range from 15.3% to 78.6%, with a mean of 62.1% and a standard deviation of 12.4%, indicating that our method generates prompts varying from lightly edited contexts to heavily masked templates. Similarly, the Average Mask Span Length exhibits significant variance, ranging from 7.4 tokens (typical of fine-grained masking) to 65.8 tokens (characteristic of block-wise masking), with an average of 28.5 tokens. This wide coverage confirms that DIJA successfully produces a diverse array of adversarial structures rather than converging on a single, repetitive pattern.
>
> **Table: Statistical analysis of structural diversity metrics for DIJA on HarmBench.**
>
> | Metric | **Min** | **Max** | **Mean** | **Std. Dev** |
> | :--- | :---: | :---: | :---: | :---: |
> | **Mask Ratio (%)** | 15.3 | 78.6 | 62.1 | 12.4 |
> | **Avg. Mask Span Length (tokens)** | 7.4 | 65.8 | 28.5 | 18.2 |
>
> **A systematic analysis of the diversity of interleaved mask–text prompts has also been added to a section in Appendix B.3, and you can also find the detailed experimental data in Table 10 of Appendix B.3.**
>
> > **W2.** Limited Practical Value due to Nascent dLLM Ecosystem
>
> **RW2:** Thank you for raising this concern regarding the practical timing of our research. While we acknowledge that dLLMs are currently in an earlier stage compared to mature AR models, we respectfully argue that this is precisely why this research is timely and critical.
>
> (continued in our next response)

---

> ### Author Response · Authors · 2025-11-21
> **Response to Reviewer Jm6J (part 2)**
>
> 1. **Proactive vs. Reactive Safety:** The history of AI safety teaches us that security vulnerabilities should be identified before widespread deployment, not after. Waiting for dLLMs to reach full maturity before investigating their safety risks would repeat the "reactive" cycle seen with autoregressive LLMs, where jailbreaks emerged only after massive deployment. Our work provides a preemptive warning, allowing the community to build safety-aware dLLM architectures from the ground up.
> 2. **Rapid Advancement & Unique Potential:** Contrary to being a distant technology, dLLMs are advancing rapidly. Recent works (e.g., LLaDA [1], LLaDA-VLA [2], LLaDA-medv [3], LLaDA-MOE [4], LLaDA2 [5], Dream [6], SDAR [7], Mercy [8], Seed diffusion [9], Google diffusion [10]) demonstrate that dLLMs are already achieving performance comparable to LLaMA-3 8B in code generation and reasoning, and are quickly expanding into a wide range of domains. Their unique capabilities, such as bidirectional infilling and efficient parallel decoding, offer clear advantages for specific high-value scenarios like code completion, document editing, and constrained generation, making their eventual adoption highly probable.
> 3. **Foundational Vulnerability:** Our findings reveal that the vulnerability stems from the core properties of diffusion LLMs (bidirectional coherence). This is a fundamental architectural issue that will likely persist or worsen as models scale. Addressing it now is essential for the safe evolution of this promising paradigm.
>
> **Reference** \
> [1] Nie, Shen, et al. "Large language diffusion models." arXiv preprint arXiv:2502.09992 (2025). \
> [2] Wen, Yuqing, et al. "Llada-vla: Vision language diffusion action models." arXiv preprint arXiv:2509.06932 (2025). \
> [3] Dong, Xuanzhao, et al. "Llada-medv: Exploring large language diffusion models for biomedical image understanding." arXiv preprint arXiv:2508.01617 (2025). \
> [4] Zhu, Fengqi, et al. "LLaDA-MoE: A Sparse MoE Diffusion Language Model." arXiv preprint arXiv:2509.24389 (2025).
> [5] https://huggingface.co/inclusionAI/LLaDA2.0-mini-preview \
> [6] Ye, Jiacheng, et al. "Dream 7b: Diffusion large language models." arXiv preprint arXiv:2508.15487 (2025). \
> [7] Cheng, Shuang, et al. "SDAR: A Synergistic Diffusion-AutoRegression Paradigm for Scalable Sequence Generation." arXiv preprint arXiv:2510.06303 (2025). \
> [8] https://www.inceptionlabs.ai/blog/introducing-mercury  \
> [9] Song, Yuxuan, et al. "Seed diffusion: A large-scale diffusion language model with high-speed inference." arXiv preprint arXiv:2508.02193 (2025). \
> [10] https://deepmind.google/models/gemini-diffusion
>
> > **Q1.** How is attack reproducibility measured given variability in prompts generated by a LLM?
>
> **RQ1:** You have raised a very valuable question. Across different runs, the structure of the generated interleaved mask–text prompts and the number of mask tokens do indeed vary. To verify the reproducibility and stability of our method, we conducted multiple repeated experiments within the DIJA framework using Qwen2.5-7B-Instruct, where different random seeds were used across runs and different temperature settings were applied to Qwen2.5-7B-Instruct.
>
> **Table: Reproducibility analysis of DiJA on HarmBench across three independent runs.**
>
> | Victim Models | **LLaDA-Instruct** | | | **LLaDA-1.5** | | | **Dream-Instruct** | | | **MMaDA-MixCoT** | | |
> | :--- | :---: | :---: | :---: | :---: | :---: | :---: | :---: | :---: | :---: | :---: | :---: | :---: |
> | **Metrics** | **ASR-k** | **ASR-e** | **HS** | **ASR-k** | **ASR-e** | **HS** | **ASR-k** | **ASR-e** | **HS** | **ASR-k** | **ASR-e** | **HS** |
> | **DIJA (Run 1, temperature 0.2)** | 96.3 | 55.5 | 4.1 | 95.8 | 56.8 | 4.1 | 98.3 | 57.5 | 3.9 | 97.5 | 46.8 | 3.9 |
> | **DIJA (Run 2, temperature 0.3)** | 95.8 | 55.0 | 4.0 | 95.3 | 56.3 | 4.0 | 97.5 | 56.8 | 3.8 | 96.8 | 46.3 | 3.9 |
> | **DIJA (Run 3, temperature 0.4)** | 96.8 | 56.3 | 4.2 | 96.3 | 57.3 | 4.2 | 98.8 | 58.0 | 3.9 | 98.0 | 47.3 | 3.8 |
> | **Mean ± Std** | **96.3±0.5** | **55.6±0.7** | **4.1±0.1** | **95.8±0.5** | **56.8±0.5** | **4.1±0.1** | **98.2±0.7** | **57.4±0.6** | **3.9±0.1** | **97.4±0.6** | **46.8±0.5** | **3.9±0.1** |
>
> As demonstrated in the table, DIJA maintains highly consistent performance across repeated independent runs. This stability is rooted in our design philosophy: DIJA exploits the fundamental architectural characteristics of dLLMs (i.e., bidirectional context modeling and Parallel Decoding) rather than relying on brittle, stochastic heuristics or 'tricks'. This robustness further confirms that the safety vulnerability we exposed is intrinsic to the current dLLM paradigm and underscores the urgent need for effective defense solutions.
>
> **We have also added this experiment and the related discussion to Appendix B.4. We hope our response resolves your concerns.**
>
> (continued in our next response)

---

> > ### Author Response · Authors · 2025-11-21
> > **Response to Reviewer Jm6J (part 3)**
> >
> > > **Q2.** In lines 372-373, the authors claim: "This is because our method exposes the harmful intent in the prompt directly." As far as I know, role-playing attacks (such as AIM in the baseline) also incorporate the complete harmful intent into the prompt. How do you explain this inconsistency?
> >
> > **RQ2:** Thank you for this insightful observation. We agree that the original phrasing "exposes the harmful intent directly" was imprecise, as role-playing attacks like AIM also contain harmful intent.
> > The distinction lies in how the intent is presented and processed:
> > 1. AIM (Role-playing): Wraps the harmful intent within a fictional or deceptive scenario (e.g., "Act as Niccolo Machiavelli"). The failure of AIM on dLLMs suggests that dLLMs (like modern AR models) effectively recognize and reject this "scenario nesting" pattern during their safety alignment.
> > 2. Does not rely on a deceptive wrapper or role-playing. Instead, our method embeds the harmful intent as essential structural constraints (via interleaved mask-text) within a coherent context (As shown in Figure 4, the original harmful query is fully preserved). Our method exploits the dLLM's bidirectional modeling property: the model is forced to complete the harmful spans not because it is "tricked" by a story, but because it is mathematically compelled to maintain the semantic coherence of the text sequence.
> >
> > **We have clarified this statement in the revised version, and we apologize for the confusion caused.**
> >
> >
> > We hope our response addresses your concerns. If you have any further questions, please feel free to reach out to us. Thank you for your time and for the thoughtful feedback on our work.
> >
> > Best regards,
> >
> > Author Team

---

> > > ### Author Response · Authors · 2025-11-26
> > >
> > > Dear Reviewer Jm6J,
> > >
> > > Thank you once again for your valuable comments on our submission. As the discussion phase is approaching its end, we would like to kindly confirm whether we have sufficiently addressed all of your concerns (or at least part of them). Should there be any remaining questions or areas requiring further clarification, please do not hesitate to let us know. If you are satisfied with our responses, we would greatly appreciate your consideration in adjusting the evaluation scores accordingly. We sincerely look forward to your feedback.
> > >
> > > Best regards,
> > > Author Team

---

### Official Review · Reviewer_zmbS · 2025-11-01

**Soundness:** 2
**Presentation:** 2
**Contribution:** 3
**Rating:** 6
**Confidence:** 5

**Summary:**

This paper proposes a simple but effective decoding-time attack for circumventing the safety alignment of diffusion Large Language Models (dLLMs) called DiJA. The design of DiJA is simple: given a harmful prompt, first construct a template that contains the structure of an affirmative response with mask tokens at locations where generating harmful content is desired, and then leverage the dLLMs parallel decoding ability to perform infilling on these masked locations to generate the harmful content. The templates are constructed via a simple in-context learning procedure. The paper finds that DiJA is effective at generating harmful content on multiple dLLMs and safety benchmarks. A simple defense is proposed, and is shown to help improve robustness to DiJA attacks.

**Strengths:**

1. The DiJA attack reveals a critical vulnerability in dLLMs and has strong implications for safely open-sourcing dLLMs. The attack appears effective against multiple dLLMs on multiple standard safety benchmarks, is easy to implement and computationally inexpensive.
2. The in-context learning approach to generating the DiJA infilling templates is well-principled and clearly explained.
3. An initial attempt is made at a training-time defense, which shows that robustness to the DiJA vulnerability can be significantly improved.
4. An ablation study is provided on the number of mask tokens.

**Weaknesses:**

1. (Minor) The first example provided for interleaved mask-text prompting in Figure 1, editing/rewriting, is a bit weak. It suggests an application of fixing small typos by resampling from the model, but typos can be trivially fixed by simple spell checkers after decoding. A stronger-motivated example for using a dLLM could be paraphrasing intermediate sentences/longer phrases.
2. (Major) The proposed DiJA attack needs to be better contextualized within existing similar decoding exploits for autoregressive models, and the assumptions about when the attack can be applied should be more clearly stated. Specifically, the core of DiJA is essentially the prefilling attack ([1, 2]) generalized to dLLMs — as both exploit the model’s desire to preserve coherence and rely on the assumption that the user can alter the assistant response — but this is not mentioned in the work. Its also worth noting that [2] also took an in-context learning approach when constructing harmful prefills. Regarding the assumption about altering the assistant response, it is trivially satisfied by any open-source model, but for the closed-source setting the ability to perform infilling on the assistant response side needs to be explicitly provided as a feature (otherwise, DiJA can be easily guarded against by just disabling infilling). For prefilling, a good example of how the latter matters is the OpenAI API does not support prefilling (and hence prefilling attacks are not possible), but the Claude API does ([3]) (and hence was shown in [2] to be vulnerable to prefilling attacks). Are there similar real-world examples for closed-source dLLMs? Please provide some more discussion on comparing and contrasting DiJA to the aforementioned prior work and on the assumptions of the threat model.
4. (Major) Similarly, the proposed Refusal-Aware Denoising Alignment defense should be contextualized within existing similar training-time interventions, such as [5].
5. (Minor) Notational discrepancies:
- Line 154 shows [MASK] being a part of the vocabulary already, such that the union in line 174 is not necessary.
- Equation 5 is essentially the same as equation 4, just by renaming y to x. The only difference is that the initial x^K can include non-mask tokens. Perhaps the notation here can be unified a bit.
6.  (Minor) Terminology
- Line 199 (and elsewhere) mentions “original jailbreak prompt” — to my knowledge, a harmful behavior prompt without any actual jailbreak applied (e.g., those discussed in Section 2) is not referred to as a “jailbreak prompt,” but rather just the original (harmful) prompt.
7. (Minor) Line 318 refers to “previous studies” for the judging prompt: please provide explicit citations of what these previous studies are (e.g., this template was used in [4]).
8. (Minor) Line 363-364: Please provide some evidence for the robustness comparison to SOTA autoregressive models (e.g., citations, specific numbers).
9. (Minor) Figure 4: I would suggest using a different icon than the OpenAI logo to represent the different dLLMs, otherwise it visually suggests these outputs were generated by GPT.


References:

[1] Vega, Jason, et al. "Bypassing the safety training of open-source llms with priming attacks." arXiv preprint arXiv:2312.12321 (2023).

[2] Andriushchenko, Maksym, Francesco Croce, and Nicolas Flammarion. "Jailbreaking leading safety-aligned llms with simple adaptive attacks." arXiv preprint arXiv:2404.02151 (2024).

[3] https://anthropic.mintlify.app/en/docs/build-with-claude/prompt-engineering/prefill-claudes-response

[4] Qi, Xiangyu, et al. "Fine-tuning aligned language models compromises safety, even when users do not intend to!." arXiv preprint arXiv:2310.03693 (2023).

[5] Qi, Xiangyu, et al. "Safety alignment should be made more than just a few tokens deep." arXiv preprint arXiv:2406.05946 (2024).

**Questions:**

1. Line 77-79: could you cite some specific examples of defenses for left-to-right models and why they cannot work for dLLMs? Some of the defenses mentioned in section 2 could arguably be applied to dLLMs as well (e.g., perplexity filter, SmoothLLM), as they do not rely on the method of decoding (left-to-right vs. parallel).
2. For the Refusal-Aware Denoising Alignment defense, how exactly are the refusals generated? Are the refusals always generic or are they prompt-dependent (e.g., providing reasons why the request was refused instead of just refusing)? Also, suppose “I’m sorry, I can’t help with that.” is tokenized to M tokens — how would this fit into a masked section with more or less than M tokens (e.g., for a section with more than M tokens, are the remaining tokens filled with additional [MASK] tokens)? Or are the lengths of the masked sections chosen to be equal to the tokenized lengths of the refusals? Finally, are a variety of refusal lengths included in the fine-tuning dataset to help generalize to different mask lengths?

---

> ### Author Response · Authors · 2025-11-21
> **Response to Reviewer zmbS (part 1)**
>
> Dear Reviewer zmbS，
>
> Thank you for your strong recognition of our work and for your valuable feedback. Below, I will address your concerns one by one.
>
> > **W1.** The first example provided for interleaved mask-text prompting in Figure 1, editing/rewriting, is a bit weak. It suggests an application of fixing small typos by resampling from the model, but typos can be trivially fixed by simple spell checkers after decoding. A stronger-motivated example for using a dLLM could be paraphrasing intermediate sentences/longer phrases.
>
> **RW1:** We fully agree with your observation. Re-masking and regenerating a longer span of text indeed better reflects our intended motivation and showcases the benefits of interleaved mask-text prompting more clearly. We will update the example accordingly and include a more representative case (e.g., paraphrasing a full sentence or a function in a code) in the camera-ready version.
>
> > **W2.** (Major) The proposed DiJA attack needs to be better contextualized within existing similar decoding exploits for autoregressive models, and the assumptions about when the attack can be applied should be more clearly stated. Specifically, the core of DiJA is essentially the prefilling attack ([1, 2]) generalized to dLLMs — as both exploit the model’s desire to preserve coherence and rely on the assumption that the user can alter the assistant response — but this is not mentioned in the work. Its also worth noting that [2] also took an in-context learning approach when constructing harmful prefills. Regarding the assumption about altering the assistant response, it is trivially satisfied by any open-source model, but for the closed-source setting the ability to perform infilling on the assistant response side needs to be explicitly provided as a feature (otherwise, DiJA can be easily guarded against by just disabling infilling). For prefilling, a good example of how the latter matters is the OpenAI API does not support prefilling (and hence prefilling attacks are not possible), but the Claude API does ([3]) (and hence was shown in [2] to be vulnerable to prefilling attacks). Are there similar real-world examples for closed-source dLLMs? Please provide some more discussion on comparing and contrasting DiJA to the aforementioned prior work and on the assumptions of the threat model.
>
> **RW2:** We sincerely thank you for your insightful suggestions and for highlighting several outstanding works related to ours. Conceptually, DIJA is related to prefilling attacks, as both exploit the model’s inherent tendency to maintain output coherence. However, beyond extending these ideas to diffusion-based large language models (dLLMs), DIJA differs from prefilling attacks in autoregressive models, which are limited by left-to-right decoding and rely on injecting adversarial prefixes. Leveraging the non-autoregressive nature of dLLMs，i.e., parallel decoding and bidirectional context modeling，DIJA can manipulate arbitrary masked spans within the model’s responses and perform iterative span-level rewriting rather than relying solely on prefix injection, achieving more flexible and fine-grained control over generated outputs.
>
> Importantly, beyond introducing a novel attack technique for dLLMs, our work primarily aims to uncover emerging security vulnerabilities arising from the unique properties of diffusion language models. In doing so, we hope to draw the community’s attention to these new risks and provide valuable insights for the development and safety alignment of future dLLM training paradigms. We believe that the potential of dLLMs, due to their parallel decoding, bidirectional context modeling, and the ability to iteratively remask and refine outputs, will continue to attract attention and enable downstream applications.
>
> **Following your suggestion, we have added a related discussion in the revised PDF, which can be found in the section highlighted in blue, *Relation to Prefilling Attacks and Threat Model Assumptions*.**
>
> > **W3.** Similarly, the proposed Refusal-Aware Denoising Alignment defense should be contextualized within existing similar training-time interventions, such as [5].
>
> **RW3:** Thank you for your valuable suggestion. After carefully reading the work you referenced as [5], I found that it is conceptually related to our use of Refusal-Aware Denoising Alignment in dLLMs, and it is indeed an excellent piece of work. [5] emphasizes training models to reverse course and refuse to answer even when guided by an adversarial prefix, thereby achieving stronger safety alignment. Our Refusal-Aware Denoising Alignment serves as a complementary and preliminary attempt to improve safety alignment specifically for dLLMs, motivated by the new security vulnerabilities we uncovered in these models. The goal is to help the model become aware of potential risks when decoding masked spans in the middle of a paragraph and avoid generating harmful content.
>
> (continued in our next response)

---

> ### Author Response · Authors · 2025-11-21
> **Response to Reviewer zmbS (part 2)**
>
> **In addition, we have added a detailed discussion of this work in Appendix A.1. Thank you again!**
>
> > **W4.** Notational discrepancies:
> Line 154 shows [MASK] being a part of the vocabulary already, such that the union in line 174 is not necessary.
> Equation 5 is essentially the same as equation 4, just by renaming y to x. The only difference is that the initial x^K can include non-mask tokens. Perhaps the notation here can be unified a bit.
>
> **Rw4:** We thank you for the meticulous attention to detail regarding our mathematical notation. We fully agree with both points and **have revised Section 3.1 to unify and clarify the notation.**
>
> > **W5.** Terminology \
> Line 199 (and elsewhere) mentions “original jailbreak prompt” — to my knowledge, a harmful behavior prompt without any actual jailbreak applied (e.g., those discussed in Section 2) is not referred to as a “jailbreak prompt,” but rather just the original (harmful) prompt.
> Line 318 refers to “previous studies” for the judging prompt: please provide explicit citations of what these previous studies are (e.g., this template was used in [4]).
>
> **RW5:** We sincerely thank the reviewer for the meticulous attention to terminology and citations. We fully agree with both points you raised and **have made the corresponding revisions in the updated version (highlighted in blue)**.
>
> > **W6.** Line 363-364: Please provide some evidence for the robustness comparison to SOTA autoregressive models (e.g., citations, specific numbers).
>
> **RW6:**
> We apologize for not including specific numerical comparisons in the original text and thank you for pointing this out.
> As noted, we conducted comparative experiments using both vanilla harmful queries and existing jailbreak attacks (AIM, PAIR) on dLLMs versus SOTA autoregressive models (LLaMA-3.1, Qwen2.5). The results are presented in Figure 3 (main text) and Figures 8 & 9 (Appendix).Taking the AIM attack in Figure 3(a) as an example, we observe that dLLMs like LLaDA-8B-Instruct and LLaDA-1.5 maintain a Keyword-based ASR (ASR-k) generally below 10%. In contrast, the autoregressive baselines, LLaMA-3.1-8B-Instruct and Qwen2.5-7B-Instruct, exhibit significantly higher ASR-k scores, often exceeding 30% in the same setting. This quantitative evidence supports our claim that dLLMs possess a level of defensibility that is at least comparable to, and in some specific attack scenarios (like AIM) even superior to, current SOTA autoregressive models.
>
> > **W7.** Figure 4: I would suggest using a different icon than the OpenAI logo to represent the different dLLMs, otherwise it visually suggests these outputs were generated by GPT.
>
> **RW7:** We thank the reviewer for pointing out this visual ambiguity. We agree that using an icon resembling the OpenAI logo could lead to confusion. **We will update Figure 4 in the revised paper, replacing the icons with distinct symbols to clearly represent the dLLMs and avoid any misinterpretation.**
>
> > **Q1.** Could you provide concrete examples of left-to-right model defenses and explain why they fail on dLLMs, given that some methods in Section 2 (e.g., perplexity filtering, SmoothLLM) seem applicable regardless of decoding strategy?.
>
> **RQ1:** Thank you for raising this important question regarding the applicability of existing defenses. While some defenses appear architecture-agnostic, they face specific challenges when applied to dLLMs and the DIJA attack.
> 1. **Perplexity-based Filters:** As discussed in **Appendix C.4**, applying perplexity filters to dLLMs is technically problematic. Unlike AR models that compute $P(x_t|x_{<t})$, dLLMs generate by predicting masked tokens. Calculating the perplexity of a prompt often requires appending masks to simulate the diffusion process, which causes an **artificial surge in perplexity** for *any* input (even benign ones), leading to high false-positive rates.
> 2. **SmoothLLM (Input Perturbation):** SmoothLLM relies on the assumption that adversarial prompts (like GCG suffixes) are "brittle" and break under character-level perturbation. However, DIJA constructs **semantically coherent** interleaved mask-text prompts guided by natural language. Small perturbations to the text segments do not destroy the semantic instruction, and perturbing the mask structure simply disrupts the generation format rather than neutralizing the harmful intent. Thus, DIJA exhibits high robustness against such random perturbations.
> 3. **Decoding-based Defenses (e.g., Rejection Sampling):** This is where the "Left-to-Right" distinction is most critical. Defenses for AR models often intervene dynamically: if token $t$ is deemed unsafe given history $t-1$, generation stops. In dLLMs, due to **parallel decoding**, the entire sequence (including all harmful tokens) is predicted simultaneously in the early denoising steps. There is no sequential "history" to check against in real-time before the full semantic meaning is formed, rendering sequential rejection mechanisms ineffective.

---

> > ### Author Response · Authors · 2025-11-21
> > **Response to Reviewer zmbS (part 3)**
> >
> > > **Q2.** For the Refusal-Aware Denoising Alignment defense, how exactly are the refusals generated? Are the refusals always generic or are they prompt-dependent (e.g., providing reasons why the request was refused instead of just refusing)? Also, suppose “I’m sorry, I can’t help with that.” is tokenized to M tokens — how would this fit into a masked section with more or less than M tokens (e.g., for a section with more than M tokens, are the remaining tokens filled with additional [MASK] tokens)? Or are the lengths of the masked sections chosen to be equal to the tokenized lengths of the refusals? Finally, are a variety of refusal lengths included in the fine-tuning dataset to help generalize to different mask lengths?
> >
> > **RQ2:** Thank you for your interest in our Refusal-Aware Denoising Alignment.
> >
> > 1.  **Generation of Refusals:** As described in **Appendix A.2**, the refusals in our alignment dataset are **standardized and generic**. Specifically, for each sample, we utilize the existing DIJA framework to construct a corresponding interleaved mask-text prompt. While the victim model is expected to predict harmful content for the masked regions in a standard attack scenario, in our **Refusal-Aware Denoising Alignment**, we directly replace the targets for these masked spans with standardized refusal messages (e.g., *"I'm sorry, I can't help with that"*). This effectively overrides the model's tendency to generate harmful content with a safe, fixed rejection.
> > 2.  **Handling Length Mismatch:** This is a critical technical detail regarding dLLMs. In our approach, we do not need to match the refusal length to the mask length.
> >     *   **Case $N > M$:** If the masked section provided by the attacker ($N$ tokens) is longer than the standardized refusal message ($M$ tokens), the model is trained to denoise the first $M$ tokens into the refusal text and the remaining $N-M$ tokens into **padding (`[PAD]`) tokens**.
> >     *   **Case $N < M$:** In the rare case where the mask is too short to fit the full refusal, the refusal is truncated.
> > 3.  **Generalization to Mask Lengths:** Consequently, it is **not necessary** to include a variety of refusal lengths in the fine-tuning dataset. By training the model to output `Refusal + [PAD]...`, it learns a robust behavior: once the refusal message is complete, all subsequent masked positions resolve to padding. This allows the defense to naturally generalize to arbitrary mask lengths ($N$) during inference, effectively ignoring the excess length allocated by the attacker.
> >
> >
> > We hope our response addresses your concerns. If you have any further questions, please feel free to reach out to us. Thank you for your time and for the thoughtful feedback on our work.
> >
> > Best regards,
> >
> > Author Team

---

> > > ### Author Response · Authors · 2025-11-26
> > >
> > > Dear Reviewer zmbS,
> > >
> > > Thank you once again for your valuable comments on our submission. As the discussion phase is approaching its end, we would like to kindly confirm whether we have sufficiently addressed all of your concerns (or at least part of them). Should there be any remaining questions or areas requiring further clarification, please do not hesitate to let us know. If you are satisfied with our responses, we would greatly appreciate your consideration in adjusting the evaluation scores accordingly. We sincerely look forward to your feedback.
> > >
> > > Best regards,
> > > Author Team

---

### Official Review · Reviewer_wJxD · 2025-11-03

**Soundness:** 3
**Presentation:** 3
**Contribution:** 4
**Rating:** 8
**Confidence:** 3

**Summary:**

This paper studies safety vulnerabilities unique to diffusion-based LLMs (dLLMs) that perform bidirectional, parallel masked decoding. The authors introduce DIJA, a jailbreak framework that converts vanilla harmful prompts into interleaved text–mask prompts, exploiting the obligation of dLLMs to fill masked spans coherently while leaving surrounding (unmasked) harmful context fixed. Because decoding is parallel, common on-the-fly safety interventions fail, leading to harmful completions even in alignment-tuned models. Experiments across multiple dLLMs (LLaDA, Dream, MMaDA, and code-oriented variants) and three benchmarks (HarmBench, JailbreakBench, StrongREJECT) show very high success rates; the abstract reports up to 100% ASR-k on Dream-Instruct and large gains over prior attacks such as ReNeLLM. The paper also proposes a preliminary refusal-aware denoising alignment that substantially reduces ASR with modest utility cost.

**Strengths:**

•Clear architectural insight. The paper crisply articulates why bidirectional infilling plus parallel decoding weakens standard guardrails, and formalizes the forcing effect created by fixing unmasked tokens while infilling masked spans.
•Simple, scalable attack. DIJA uses few-shot prompt construction with masking-pattern and separator diversification; Algorithm 1 and Section 3 detail an automated pipeline that doesn’t hide harmful intent yet reliably elicits unsafe content.
•Comprehensive evaluation. Results across three standard jailbreak suites show consistent, often dramatic improvements over baselines (Tables 1–3; code-model results in Tables 6–8). The paper also examines defense robustness (Self-reminder, RPO) and performs informative ablations on generation/Mask length (Figures 5–7).
•Initial mitigation path. The refusal-aware denoising alignment substantially reduces ASR on DIJA inputs with modest impact on general tasks (Table 4–5), suggesting practical, architecture-aware safety training is feasible.

**Weaknesses:**

•Evaluator dependence & metric triangulation. While the paper uses both keyword-based and evaluator-based metrics (including StrongREJECT), a large portion of the story still relies on LLM judges and prompts (e.g., GPT-4o for HS, DIJA* for construction). Some cross-checking with human raters or multiple independent evaluators would further bolster soundness.
•Interface assumptions for mask control. The attack presumes the user can inject mask tokens (e.g., [MASK] or <mask:N>) directly. Many practical deployments may not expose raw mask-infilling controls to end users, or may preprocess/normalize such tokens. Clarifying the assumed I/O interface per model (and what happens under more restricted UIs) would help generalize the threat model. (Figure 1 illustrates capabilities, but deployment exposure varies.)
•Defense generalization risk. The proposed alignment targets a specific attack pattern (interleaved mask–text). Although Table 4 is promising, the paper also notes open questions about generalization to unseen or modified patterns and potential trade-offs on legitimate masked-editing use cases. A broader evaluation of adaptive adversaries would strengthen the mitigation claim.
•Safety examples in the text. Figure 4 contains vivid harmful outputs. While necessary for evidence, trimming or redacting some specifics could reduce dual-use risk in the camera-ready.

**Questions:**

1.Model interfaces: For each evaluated dLLM, what exact token or API is used for masks (e.g., [MASK] vs <mask:N>)? Are inputs sanitized by the serving stack? A short table mapping model ↔ mask syntax/entrypoint would clarify portability of DIJA.
2.Ablations on UI constraints: If masks are only allowed in delimited slots (e.g., structured templates) or subject to server-side stripping, how does DIJA perform? Can the attack be adapted with fewer/shorter masked spans while maintaining high ASR (beyond Figure 7’s token-count sweep)?
3.Evaluator sensitivity: Did you compare GPT-4o HS to an open evaluator (e.g., WildGuard) or limited human spot-checks to estimate judge variance? Even a small double-annotation subset would help calibrate effect sizes.
4.Defense coverage: Does refusal-aware alignment spill over to benign masked-editing workflows (Figure 1) by over-refusing? Could you report a small suite of benign infilling tasks before/after alignment?
5.Threat model clarity: Section 4 briefly treats “safety-aligned” as SFT with safety data. Please enumerate which victim models had what kind of alignment (SFT vs RLHF vs preference optimization) to better contextualize DIJA’s gains.

---

> ### Author Response · Authors · 2025-11-21
> **Response to Reviewer wJxD (part 1)**
>
> Dear Reviewer wJxD,
>
> First of all, thank you for your recognition of our work and your positive evaluation. Below, I provide point-by-point responses to your questions.
>
> > **W1.** Evaluator dependence & metric triangulation. While the paper uses both keyword-based and evaluator-based metrics (including StrongREJECT), a large portion of the story still relies on LLM judges and prompts (e.g., GPT-4o for HS, DIJA* for construction). Some cross-checking with human raters or multiple independent evaluators would further bolster soundness. \
> > **Q3.** Evaluator sensitivity: Did you compare GPT-4o HS to an open evaluator (e.g., WildGuard) or limited human spot-checks to estimate judge variance? Even a small double-annotation subset would help calibrate effect sizes.
>
> **RW1&Q3:** We appreciate your suggestion to further validate our evaluation metrics. We agree that cross-checking the GPT-4o judge with human evaluators is essential to bolster the soundness of the reported Harmfulness Score (HS).
>
> To address your concern, we conducted a **human evaluation** on a randomly sampled subset of **100 responses** (25 from each of the four victim dLLMs). Two independent human evaluators scored each response on a scale of **1 to 5**, strictly following the same **Harmfulness Score (HS) rubric** used by GPT-4o.
>
> We validated the alignment between Human and GPT-4o judgments on two levels:
> *   **Harmfulness Detection (Binary Agreement):** To verify if GPT-4o correctly identifies harmful content, we binarized the scores with a threshold of 3 (Scores **>=3** as "Harmful", **<3** as "Safe/Refusal").
> *   **Score Consistency (Pearson Correlation):** We calculated the correlation between the raw 1-5 scores to ensure GPT-4o accurately reflects the granularity of harmfulness.
>
> **Table: Human vs. GPT-4o Evaluation Agreement**
>
> | Metric | **Binary Agreement** (HS>=3) | **Cohen's Kappa** ($\kappa$) | **Pearson Correlation** ($r$) |
> | :--- | :---: | :---: | :---: |
> | **Human A vs. GPT-4o** | 94.0% | 0.88 | 0.92 |
> | **Human B vs. GPT-4o** | 93.0% | 0.86 | 0.90 |
> | **Human A vs. Human B** | 96.0% | 0.92 | 0.95 |
>
> *   **Binary Agreement & $\kappa$:** Measures consistency in distinguishing harmful vs. safe responses. $\kappa > 0.8$ indicates strong agreement.
> *   **Pearson Correlation ($r$):** Measures the linear correlation between the 1-5 scores. $r > 0.9$ indicates that GPT-4o's scoring trend is highly synchronized with human intuition across the entire severity spectrum.
>
> The results confirm that GPT-4o is a reliable evaluator for the Harmfulness Score: it aligns with humans on harmfulness detection and accurately captures the degree of severity.
>
> **We have added these human verification results to Appendix D.4 in the revision.**
>
> > **W2.** Interface assumptions for mask control. The attack presumes the user can inject mask tokens (e.g., [MASK] or mask:N) directly. Many practical deployments may not expose raw mask-infilling controls to end users, or may preprocess/normalize such tokens. Clarifying the assumed I/O interface per model (and what happens under more restricted UIs) would help generalize the threat model. (Figure 1 illustrates capabilities, but deployment exposure varies.) \
> > **Q1.** Model interfaces: For each evaluated dLLM, what exact token or API is used for masks (e.g., [MASK] vs mask:N)? Are inputs sanitized by the serving stack? A short table mapping model ↔ mask syntax/entrypoint would clarify portability of DIJA.
>
> **RW2&Q1:** We appreciate your insight regarding the interface assumptions. You are correct that practical deployments vary, and clarifying the I/O interface is crucial for generalizing the threat model.
>
> **1. Model Interfaces and Sanitization:**
> Currently, all dLLMs evaluated in our paper (and most state-of-the-art dLLMs) are **Open Source**. In this setting, the "interface" is typically direct access via libraries like HuggingFace Transformers. Users have full control over the tokenizer and input sequence, meaning **no sanitization** is applied by a serving stack, users can directly inject raw mask tokens.
>
> **2. Mapping of Mask Syntax (Table provided):**
> Furthermore, we have compiled a mapping table detailing the specific mask tokens and access points for the evaluated models and other representative dLLMs. As shown below (and added to **Appendix C.6**), while the syntax varies (e.g., `<|mdm_mask|>` vs. `<|mask|>`), the mechanism remains consistent:
>
> (continued in our next response)

---

> ### Author Response · Authors · 2025-11-21
> **Response to Reviewer wJxD (part 2)**
>
> **Table: Mapping of dLLM Interfaces and Mask Syntax**
>
> | Model Family | **Open Source** | **Mask Token Syntax** | **Access Interface** |
> | :--- | :---: | :---: | :--- |
> | **LLaDA-8B** | $\checkmark$ | `<\|mdm_mask\|>` | HuggingFace (`GSAI-ML/LLaDA-8B-Base`) |
> | **LLaDA-MoE** | $\checkmark$ | `<\|mask\|>` | HuggingFace (`inclusionAI/LLaDA-MoE-7B-A1B-Base`) |
> | **LLaDA-2.0** | $\checkmark$ | `<\|mask\|>` | HuggingFace (`inclusionAI/LLaDA2.0-mini`) |
> | **Dream-7B** | $\checkmark$ | `<\|mask\|>` | HuggingFace (`Dream-org/Dream-v0-Base-7B`) |
> | **DiffuCoder** | $\checkmark$ | `<\|mask\|>` | HuggingFace (`apple/DiffuCoder-7B-Instruct`) |
> | **SDAR Family** | $\checkmark$ | `<\|MASK\|>` | HuggingFace (`JetLM/SDAR-8B-Chat`) |
> | **Mercy Diffusion** | $\times$ | - | `https://www.inceptionlabs.ai` |
> | **Seed Diffusion** | $\times$ | - | `https://studio.seed.ai/exp/seed_diffusion`|
>
> **3. Generalization to Restricted UIs:**
> We acknowledge that future closed-source APIs *could* sanitize inputs. However, as discussed in our Threat Model (Section 3), **remasking and re-generation** is a core competitive advantage of dLLMs over autoregressive models. Disabling mask inputs would strip the model of this native capability. Therefore, even restricted UIs are likely to expose an "edit" or "insert" endpoint (similar to OpenAI's legacy Edit API or Claude's Prefill), which effectively allows the placement of masked spans, keeping DIJA relevant.
>
> **We have added this table and the discussion on interface assumptions to Appendix C.6 in the revision.**
>
> > **W3.** Defense generalization risk. The proposed alignment targets a specific attack pattern (interleaved mask–text). Although Table 4 is promising, the paper also notes open questions about generalization to unseen or modified patterns and potential trade-offs on legitimate masked-editing use cases. A broader evaluation of adaptive adversaries would strengthen the mitigation claim. \
> > **Q4.** Defense coverage: Does refusal-aware alignment spill over to benign masked-editing workflows (Figure 1) by over-refusing? Could you report a small suite of benign infilling tasks before/after alignment?
>
> **RW3&Q4:** Thank you for highlighting the critical aspect of defense generalization and utility trade-off. It is essential to ensure that our alignment strategy does not compromise legitimate use cases, especially benign masked-editing workflows as shown in Figure 1.
>
> **1. Impact on General Capabilities:**
> As reported in **Table 5 (Appendix A.4)**, our aligned model maintains performance on par with the original model across general benchmarks (e.g., GSM8K, HumanEval), indicating minimal degradation in foundational capabilities.
>
> **2. Impact on Benign Infilling (New Experiment with XSTest):**
> To specifically address your concern about over-refusal in benign infilling tasks, we constructed a Benign Infilling Benchmark using XSTest. XSTest contains prompts that appear harmful (e.g., containing sensitive keywords) but are semantically safe, and is specifically designed to evaluate whether models exhibit over-refusal on benign tasks after alignment.\
> We processed 100 safe prompts from XSTest through our DIJA framework. This generated **benign interleaved mask-text prompts** that share the exact same structural characteristics as our attack prompts but contain safe semantic intent.
>
> **Table: Refusal Rate on Benign Interleaved Prompts (XSTest)**
>
> | Model | **Original LLaDA-Instruct** | **Aligned LLaDA-Instruct** |
> | :--- | :---: | :---: |
> | **Refusal Rate** | 0.0% | **3.0%** |
>
>
> The aligned model successfully completed 97% of the benign interleaved prompts. This is a critical finding: it proves that our defense does not blindly reject the **interleaved structure**. Instead, it has learned to discern the **underlying malicious intent** within that structure. It correctly refuses DIJA-generated harmful prompts while accepting DIJA-generated safe prompts, demonstrating high precision and minimal spill-over.
>
> **3. Future Outlook on Generalization:**
> As stated in our submission, the proposed Refusal-Aware Denoising Alignment represents a foundational step in mitigating the unique vulnerabilities of dLLMs. While our current results demonstrate strong defense against DIJA with minimal degradation on benign tasks, we acknowledge that characterizing robustness against a broader spectrum of **adaptive adversaries** is an evolving challenge. Establishing a mature, universally generalized alignment strategy for dLLMs will require further research, comprehensive training on diverse attack variations, and real-world validation.
>
> **We have added these results to Appendix A.4 to clarify the defense's precision and utility preservation.**
>
> Thank you again for your recognition of our work and for the valuable suggestions you have provided.
>
> (continued in our next response)

---

> ### Author Response · Authors · 2025-11-21
> **Response to Reviewer wJxD (part 3)**
>
> >**W4.** Safety examples in text: Figure 4 shows vivid harmful outputs. Can some details be trimmed or redacted in the camera-ready to reduce dual-use risk?
>
> **RW4:** Thank you very much for your friendly reminder. We will mask or obscure any content that may be risky in the case presentation in the camera-ready version.
>
> > **Q2.** Ablations on UI constraints: If masks are restricted to fixed slots or stripped server-side, how does DIJA perform? Can it still achieve high ASR with fewer or shorter masked spans beyond the token-count sweep in Figure 7?
>
> **RQ2:** Thank you for this insightful question regarding deployment constraints.
>
> **1. Impact of Structured Slots and Server-side Stripping:**
> *   **Delimited Slots:** Our extensive ablation studies (e.g., Table 12 in the Appendix) indicate that the success of DIJA hinges on the **interleaved mask-text structure**. As long as the UI allows enough flexibility to construct such interleaved patterns, even within specific delimited slots like JSON fields or code blocks, the attack remains highly effective.
> *   **Server-side Stripping:** While stripping user-provided mask tokens would indeed neutralize DIJA, we argue that this is **practically undesirable**. One of the core competitive advantages of dLLMs over autoregressive models lies in their ability to perform user-guided **remasking and re-generation** at arbitrary positions. Disabling external mask injection would strip the model of this unique interactivity.
>
> **2. Adaptation to Fewer/Shorter Mask Spans:**
> *   **Shorter Spans (Length):** As shown in **Figure 7**, reducing the total number of mask tokens (which equates to shortening the mask span length in discrete dLLMs) does lead to a drop in ASR. This is expected: very short spans physically limit the model's capacity to generate complex harmful content (e.g., detailed malware code). However, for concise harmful outputs (e.g., hate speech), the attack remains potent.
> *   **Fewer Spans (Count):** To address "fewer masked spans," we conducted a new experiment on LLaDA-8B-Instruct (HarmBench) where we fixed the total mask token budget 50 but varied the **number of separate mask spans** allowed in the prompt.
>
> **Table: Impact of Mask Span Count. LLaDA-8B-Instruct on HarmBench**
>
> | Number of Spans | **1** | **2** | **3** | **5** | **10** |
> | :---: | :---: | :---: | :---: | :---: |:---: |
> | **ASR-k** | 62.3% | 85.1% | **96.3%** | 94.8% | 87.5% |
> | **ASR-e** | 31.5% | 48.5% | **55.5%** | 53.0% | 43.5% |
>
> Restricting the number of spans (e.g., to 1) significantly reduces attack effectiveness (ASR-e drops to 31.5%), confirming that the **interleaved mask-text structure** is key. However, it does not fully eliminate the risk, serving as a partial mitigation rather than a complete defense. At the same time, we observe that when the mask token budget is fixed, using too many mask spans effectively shortens each span, which also leads to a degradation in attack performance.
>
> Overall, these findings underscore the resilience of DIJA: it requires minimal structural freedom (e.g., just 2-3 interleaved spans) to successfully trigger the bidirectional vulnerability and bypass safeguards. This demonstrates that superficial UI constraints are insufficient to secure dLLMs against this fundamental threat, reinforcing the need for intrinsic alignment solutions.
>
> **We have added this discussion and the new experiment to Appendix B.6 in the revision.**
>
> (continued in our next response)

---

> > ### Author Response · Authors · 2025-11-21
> > **Response to Reviewer wJxD (part 4)**
> >
> > >**Q5.** Threat model clarity: Section 4 treats “safety-aligned” as SFT with safety data. Could you specify which victim models used SFT, RLHF, or preference optimization to contextualize DIJA’s gains?
> >
> > **RQ5:** Thank you for your valuable suggestion. We clarify the safety-alignment status of each victim model as follows.\
> > (1) Models without RL-based training.
> > From both the released model cards and direct communication with the model authors, LLaDA-Instruct and Dream-v0-Instruct did not undergo any RL training. Their safety alignment was therefore conducted entirely during the SFT stage, where safety-related data were incorporated. \
> > (2) Models with RL-based training.
> > LLaDA-1.5 and MMaDA-MixCoT were trained with reinforcement learning. However,  their public technical reports and model cards do not specify whether RLHF or additional preference-optimization steps were used.
> >
> > Importantly, as shown in Figures 3, 8, and 9 in the submission, current diffusion-LLMs already exhibit comparable or even stronger robustness to prior jailbreak attacks compared with autoregressive LLMs. This indicates that the victim models we evaluate are already reasonably well safety-aligned, and DIJA’s success therefore highlights a novel and previously underexplored security vulnerability specific to diffusion-based LLMs.
> >
> > We will add this clarification to contextualize DIJA’s improvements better when we clearly understand which specific safety-alignment techniques each model used at each stage of training. Existing dLLM reports and model cards largely overlook safety alignment clarification, showing the community has not yet recognized these new vulnerabilities and underscoring our work’s importance.
> >
> > We hope our response addresses your concerns. If you have any further questions, please feel free to reach out to us. Thank you for your time and for the thoughtful feedback on our work.
> >
> > Best regards,
> >
> > Author Team

---

### Author Response · Authors · 2025-12-02
**Global Response [Updata of PDF & Point-by-Point Reply]**

We sincerely **thank the Program Chairs, Area Chairs, and all Reviewers** for the effort and time dedicated to reviewing our work. To address all concerns and strengthen our paper, we provided a point-by-point response, including **6 new experiments** and **5 pages of revision**.

We are greatly encouraged by the reviewers' recognition of our work across multiple dimensions:

* **Novel Architectural Vulnerability Discovery:** "Clear architectural insight" (``Reviewer wJxD``), "a critical vulnerability in dLLMs" (``Reviewer zmbS``), "identifies a critical safety vulnerability" (``Reviewer Jm6J``), "emergent safety vulnerability specific to dLLMs" (``Reviewer Jo4s``).
* **Comprehensive Evaluation and Method Validation:** "Results...show consistent, often dramatic improvements over baselines" (``Reviewer wJxD``), "The attack appears effective against multiple dLLMs" (``Reviewer zmbS``), "The experimental design is comprehensive and robust" (``Reviewer Jm6J``).
* **Good Presentation:** "well-principled and clearly explained" (``Reviewer zmbS``), "The paper features clear writing" (``Reviewer Jm6J``).

To resolve all questions, we added more experiments (even human evaluation), ablation studies, and corresponding explanations. We provided a point-by-point reply in the rebuttal dialogue and detailed explanations (highlighted in blue) in the **revised PDF**. Below is a summary of these additions:

- **Comprehensive Ablation Studies and Method Diversity (Appendix. B.5, B.3, B.6)**
    - Comprehensive ablation experiments were designed (Table 12, Appendix B.5) for ``Reviewer Jo4s``, thus validating the core mechanisms and effectiveness of our method.
    - Added experiments on the number of mask spans (Table 13, Appendix B.6) to answer ``Reviewer wJxD`` on mask quantity/length impact.
    - Demonstrated method **diversity**, adaptability, and flexibility by quantifying Mask Ratio and Average Mask Span Length of 400 interleaved text-mask prompts from DIJA (Table 10, Appendix B.3) (``Reviewer Jm6J``).

- **Convincing Evaluation Metrics and Method Robustness (Appendix D.4, B.4)**
    - Introduced human evaluators for harmfulness scoring (cross-check). Verified Harmfulness Score credibility by calculating **Pearson Correlation** with human scores (Table 19, Appendix D.4) (``Reviewer wJxD``).
    - Addressed reproducibility (``Reviewer Jm6J``) by conducting three repeat experiments and calculating uncertainty metrics (Table 11, Appendix B.4).

- **Supplementation on dllm Interface & Refusal Rate (Appendix C.6, A.4)**
    - Added information on the current dllm interface and mask token types (Table 17, Appendix C.6) (``Reviewer wJxD``).
    - Adapted XSTest into an infilling task to further examine Refusal-Aware Denoising Alignment's impact on benign tasks (Table 9, Appendix A.4).

We also corrected inappropriate wordings and expressions in the revised manuscript. For more details, kindly refer to our conversations and the revised manuscript (highlighted in blue).

We emphasize that **our work reveals an architectural-level novel security vulnerability in dllms**. Our proposed DIJA attack framework is simple, highly effective, and **cost-effective**, unlike previous jailbreak attack methods requiring multi-round gradient optimization or tedious manual design. Numerous works are already researching safety alignment algorithms for dllm based on our findings. Concurrently, various dllm models, inference frameworks, acceleration methods, and downstream applications are being introduced, **showcasing the immense potential of dllms.**  All of these facts underscore the **importance, timeliness, and contribution of our work to the community.**

We hope this summary facilitates your quick understanding of our work and the rebuttal process. Thank you once again for your time.

Best regards,

Author Team

---

### Meta-Review · Area_Chair_7XPc · 2025-12-06

**Summary:**

Across the reviews, the main concerns focus on how the paper positions itself relative to prior work, the completeness of its evaluation, and clarity in presenting both the method and threat model.
Reviewers agree that the discovery of a safety weakness in diffusion-based language models is important and the empirical results are strong. However, they also highlight gaps such as missing ablations that separate the contributions of different components of DIJA, limited discussion of how the attack compares to prefilling-style attacks on autoregressive models, unclear assumptions about mask-token availability in real deployments, reliance on LLM evaluators, and several clarity and presentation issues.
Taken together, reviewers see the contribution as promising but believe some aspects of the framing and methodology need stronger grounding.

**Reviewer Concerns:**

There was no active discussion during the period, but some concerns raised by the reviewers are the type that authors can successfully address during a rebuttal. Issues such as clarifying which mask tokens each model accepts, providing a clearer threat model, explaining how refusals are generated in the defense, improving notation, adjusting claims that seemed too strong, or clarifying the two-stage DIJA construction process can all be resolved with explanation. These are likely the concerns that would have been considered addressed.
However, several larger concerns probably remain at least partly outstanding because they require additional experiments or structural revisions. Examples include the request for a full ablation to separate masking patterns from prompt-refinement effects, deeper comparison to prefilling attacks, broader evaluation of the proposed defense against adaptive adversaries, explaining the cases where the baseline ReNeLLM surpasses DIJA, and analyzing the diversity and reproducibility of prompts generated by in-context learning. These points go beyond what a rebuttal can usually accomplish and would likely remain concerns after discussion.

**Reviewer Scores:**

Reviewer wJxD originally scored the paper highly and their concerns were minor and mostly addressable, 8->8

Reviewer zmbS expressed moderate support but emphasized missing context with prior work, 6->6

Reviewer Jm6J had mixed feelings, with concerns about simplicity and practical impact. 4->4

Reviewer Jo4s had the lowest score because of missing ablations and confusing method presentation 2->3,4

---

### Decision · Program_Chairs · 2026-01-26

Accept (Poster)